# ERGO: Entropy-guided Resetting for Generation Optimization in Multi-turn Language Models

## Abstract

Interactive AI systems face critical reliability challenges as conversation length increases, with Large Language Models (LLMs) exhibiting significant performance degradation when deployed in extended multi-turn environments. This degradation, manifesting as reduced accuracy, decreased confidence, and a 112% increase in response variability (unreliability), represents a fundamental robustness failure in interactive machine learning systems. We introduce ERGO (**E**ntropy-guided **R**esetting for **G**eneration **O**ptimization), a principled approach to maintaining system reliability and performance in interactive environments by monitoring internal uncertainty signals and triggering automated context consolidation when degradation is detected. ERGO uses Shannon entropy over next token probability distributions as a real-time indicator of system robustness, automatically restructuring interaction history when uncertainty spikes indicate potential failure modes. Evaluated across multiple LLMs in interactive task scenarios, ERGO improves average performance by 56.6% over degraded multi-turn baselines, completely recovers the 15% drop in peak performance reliability, and reduces response variability by 35.3%. Our results demonstrate that entropy-based uncertainty monitoring provides an effective framework for building robust interactive ML systems that maintain consistent performance despite the inherent unreliability of accumulated and noisy conversational context.

## 1 Introduction

Interactive machine learning systems must maintain reliable performance across extended user engagements, yet recent research reveals a critical vulnerability. Large Language Models (LLMs) experience substantial failures in multi-turn interactive environments compared to single-turn deployments (Laban et al., 2025; Gupta et al., 2024). This degradation represents a fundamental challenge for reliable ML deployment, as interactive systems accumulate contextual "noise" that progressively corrupts their decision-making processes, leading to a 112% increase in response unreliability and significant performance drops (Laban et al., 2025).

The robustness challenges in interactive environments stem from the mismatch between training assumptions and deployment realities. While LLMs are typically trained and evaluated on clean, well-structured single-turn examples, real-world deployment involves extended interactions where context accumulates incrementally, creating distribution shift and progressive corruption of the input signal. This accumulated context acts as increasingly unreliable data, causing models to become "lost" in the interaction flow and exhibit unpredictable failure modes.

Existing approaches to maintaining reliability in interactive systems remain limited. Methods based on task classification, retrieval, or context compression lack generality and often require system-specific fine-tuning (Wu et al., 2023). More critically, these approaches fail to address the fundamental

Submitted to Reliable ML from Unreliable Data Workshop @ NeurIPS 2025

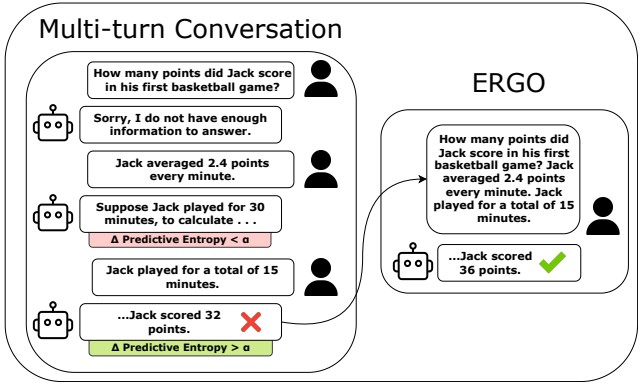

Figure 1: Illustrative comparison of a standard multi-turn conversational AI and the ERGO system

issue, detecting when the interactive system has entered an unreliable state and requires intervention to restore robustness.

We introduce ERGO (**E**ntropy-guided **R**esetting for **G**eneration **O**ptimization), the first practical framework for maintaining reliability and performance in interactive ML environments through dynamic uncertainty monitoring. ERGO addresses the core challenge of building reliable systems from unreliable interactive data by computing Shannon entropy over next-token probability distributions (Malinin and Gales, 2018; Xiao and Wang, 2022) as a real-time indicator of system robustness. When entropy spikes indicate that accumulated conversational context has become sufficiently corrupted to threaten system reliability, ERGO triggers entropy-guided context reconstruction, automatically distilling reliable information from the degraded interaction state while discarding accumulated noise that compromises robustness, a visual representation of this can be seen in Figure 1.

Empirical results across multiple interactive scenarios demonstrate ERGO's effectiveness for robust interactive ML, 56.6% improvement in average performance compared to multi-turn baselines, 24.7% increase in peak performance levels, and a 35.3% reduction in the response unreliability typically observed in extended interactions. Our approach outperforms existing strategies while providing better precision and timing compared to alternate baselines, establishing entropy-guided uncertainty monitoring as an effective framework for reliable interactive ML systems. To verify our findings and reproduce the results, please refer to the anonymized code repository found at the following link: https://anonymous.4open.science/r/ERGO-2F58

## 2 Background and Related Works

Recent work has documented substantial performance degradation in multi-turn LLM conversations. Laban et al. (2025) demonstrated that model performance rates dropped by 39% on average in multi-turn settings across six domains. Gupta et al. (2024) formalized task-switch sensitivity using probability ratios, showing how conversation history compounds model confusion. While Laban et al. (2025) tested remediation approaches and managed to improve average performance losses by 15-20%, these face substantial verbosity and practicality constraints. Agent-based frameworks (Wu et al., 2023) explore system-level solutions but do not target fundamental model limitations during generation.

### 2.1 Entropy Based Uncertainty Estimation

Entropy-based uncertainty estimation provides the theoretical basis for our method, grounding ERGO's use of internal model signals. Prior work has used predictive entropy to quantify model confidence in classification and generation tasks (Malinin and Gales, 2018; Xiao and Wang, 2022), implicitly linking internal uncertainty to external behavior. More recent approaches extend this to semantic-level uncertainty using semantic-aware entropy measures (Kuhn et al., 2023) or trainable proxies derived from hidden representations (Kossen et al., 2024). While these methods improve semantic fidelity, they often rely on sampling or auxiliary models. In contrast, we use token-level

entropy, computed directly from the model's next-token distribution, as a low-cost proxy for real-time monitoring. Unlike prior work that applies entropy primarily for evaluation or filtering, we use it as a temporal signal to detect context degradation and trigger prompt restructuring.

## 2.2 Inference-Time Interventions

Inference-time control methods intervene on frozen models by manipulating internal activations, modifying output logits, or reranking candidate outputs. For example, Li et al. (2024) introduced activation-level interventions to elicit truthful answers without fine-tuning, shifting hidden states toward truthful completions. Similarly, Turner et al. (2024) developed activation engineering techniques that steer the behavior of the model by editing intermediate representations during decoding. These methods act directly on the output path of the model and often rely on internal signal manipulation.

In contrast, our approach introduces a policy layer outside of the model that monitors uncertainty and intervenes by restructuring the user's input. We do not modify the internal computation or sampling process of the model.

## 2.3 Backtracking and Prompt Restructuring

Several recent approaches have explored controlled backtracking during generation. Cundy and Ermon (2024) augmented the decoding space with a 'backspace' action to revert low-probability generations, while Zhang et al. (2024) uses a special [RESET] token to discard unsafe prefixes. Other strategies such as Self-Refine (Madaan et al., 2023) allowed iterative refinement by prompting the model to critique and revise its own output. These methods operate on generated content and typically require multi-step decoding or auxiliary supervision.

Our intervention departs from this paradigm by focusing on upstream correction. Instead of rewriting the model's response, we update the user's prompt to recover task coherence, using rising entropy as the intervention trigger. This shifts the optimization target from output correction to input re-specification, which is more lightweight and avoids cumulative reasoning errors. To our knowledge, this is the first method that uses entropy-based signals to restructure user input mid-conversation, rather than adjusting the model's internal behavior or downstream output.

# 3 Entropy-Guided Context Resetting

## 3.1 Rise in Average Token Level Entropy

At each turn of the conversation, the average token-level entropy is calculated by measuring the uncertainty of the model's token probability distribution when generating each token in its output.

Suppose the model produces a sequence of tokens $t_1, t_2, \ldots, t_n$ at a given turn. For each token $t_i$, the model assigns a probability distribution $P_i$ over the vocabulary $V$, where $P_i(v)$ is the probability assigned to token $v \in V$ at position $i$.

The entropy at position $i$ is computed as:

$$H_i = -\sum_{v \in V} P_i(v) \log P_i(v)$$

The average token-level entropy $\bar{H}$ for the turn (covering $n$ generated tokens) is then:

$$\bar{H} = \frac{1}{n} \sum_{i=1}^{n} H_i$$

This metric quantifies the model's overall uncertainty when generating the turn. Higher $\bar{H}$ indicates greater uncertainty and a more diffuse token distribution, while the lower $\bar{H}$ indicates more confident and peaked predictions (Malinin and Gales, 2018; Xiao and Wang, 2022).

For each subsequent turn $t$ in the conversation, the change in average token-level entropy is calculated to monitor fluctuations in model uncertainty. Let $\bar{H}^{(t)}$ denote the average token-level entropy at turn $t$, as defined previously.

The change in predictive entropy between consecutive turns is defined as:

$$\Delta \bar{H}^{(t)} = \bar{H}^{(t)} - \bar{H}^{(t-1)}$$

A positive $\Delta \bar{H}^{(t)}$ indicates that the uncertainty of the model has increased relative to the previous turn.

## 3.2 Threshold-Based Trigger for Context Reset

For each model we calibrate an entropy change threshold ($\tau$). When the change in predictive entropy satisfies the following condition:

$$\Delta \bar{H}^{(t)} > \tau$$

The system deems that the uncertainty of the model is rising beyond an acceptable margin. This is interpreted as a signal that the evolving conversation context may be inducing compounding uncertainty or drift. A detailed analysis of the threshold selection process is provided in Appendix A, while an analysis of ERGO's sensitivity to entropy thresholds is provided in Appendix B.

## 3.3 Context Reset Protocol

Upon detection of $\Delta \bar{H}^{(t)} > \tau$, an automated context reset protocol is initiated. This protocol proceeds in the following steps:

**I. Prompt Rewriting:**
The user's inputs up to turn $t$ are provided to the model. The model is asked to rewrite these inputs into a single-turn, optimized prompt that preserves relevant task information while reducing ambiguity and redundancy.

**II. Isolated Generation (New Chat Simulation):**
The rewritten prompt is passed into a new instance of the model, simulating a stateless chat environment with no memory of prior turns. The model then generates a response $R_{\text{opt}}$ to this rewritten prompt.

**III. Branch Continuation:**
A new dialogue branch is created that begins from the rewritten prompt and response. This maintains continuity from the optimized state rather than the potentially degraded original context.

# 4 Experimentation Background

## 4.1 Simulation Scale & Parameters

Our simulation follows the protocol of Laban et al. (2025) with the only change being the implementation of ERGO. We evaluate a suite of five leading instruction-tuned LLMs: **Phi-4** (Abdin et al., 2024), **LLaMA 3.1-8B Instruct** (Grattafiori et al., 2024), **GPT-4o** (Hurst et al., 2024), **GPT-4.1** (OpenAI, 2025), and **GPT-4o-mini** (OpenAI, 2024). All models are used in their publicly released variants without additional fine-tuning.

Generation settings are standardized across models with temperature set to 1.0. For entropy calculations, we note an important methodological constraint: OpenAI models provide access to only the top-20 logprobs through their API. This limitation affects the precision of entropy estimates, particularly for tasks with shorter responses such as *Actions* and *Data-to-text*, where the restricted probability space may not capture the full uncertainty of the model's predictions.

We conduct 3 independent simulation runs for each dataset using 100-question samples, with the exception of the Data-to-text dataset, for which evaluations were performed on a 50-question subset

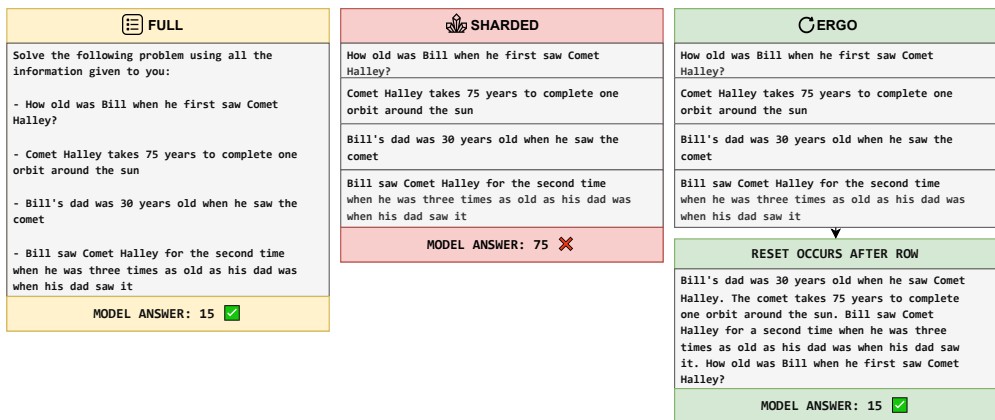

Figure 2: Example LLaMA 3.1-8B Instruct run on a GSM8K question with 🗒 FULL, 🌿 SHARDED and ↻ ERGO settings. Each row represents a separate prompt given to the model while each table represents a context window.

over 3 runs. All other experimental settings and baseline figures are adopted directly from Laban et al. (2025).

We compare three settings:

🗒 FULL: Simulates a single-turn, fully-specified conversation using the sharded instruction. The shards are combined into a single bullet-point list (one shard per line), prefaced by a directive to complete the task using all listed points. This setting serves as an upper bound for performance, providing a target for evaluating how closely multi-turn intervention methods can approximate single-turn optimality.

🌿 SHARDED: Sequential shard presentation as in the original (Laban et al., 2025) LLMs-lost-in-conversation experiment.

↻ ERGO: Our entropy-guided reset mechanism applied upon exceeding the entropy threshold.

Figure 2 provides an example of a run on each setting. This evaluation isolates the effect of ERGO relative to both single-pass and original multi-turn baselines.

## 4.2 Tasks

We evaluated models on five representative generation tasks, each framed as a multi-turn interaction over sharded instructions and augmented them with our entropy-guided context resetting method (Section 3). For each task, we used 220-325 constructed prompts from the datasets created by Laban et al. (2025). We simulate a multi-turn conversation, feeding the model one shard at a time. At each assistant turn, we compute the average token-level entropy and track its change $\Delta \bar{H}^{(t)}$. Whenever $\Delta \bar{H}^{(t)}$ exceeds the calibrated threshold $\tau$, we invoke our reset protocol - prompt rewriting, isolated regeneration, branch continuation - before continuing.

Below we briefly summarize what the assistant must do in each task:

🖥 CODE: Convert natural-language problem description into a correct Python function. Outputs are validated by executing against the reference test suite (Chen et al., 2021; Jain et al., 2024).

🗄 DATABASE: Given a database schema and a user request, generate an SQL query that returns the requested data. Correctness is checked by running the query on the Spider-derived database (Yu et al., 2018).

🖳 ACTIONS: Given API schemas plus high-level user instruction, emit valid code-style API calls that fulfill the intent. This is verified against the Berkeley Function Calling Leaderboard definitions (Yan et al., 2024).

183 📝 DATA-TO-TEXT: Take a structured data table and metadata and write a single caption that
184 highlights its key insight. Adapted from ToTTo and evaluated using BLEU (scaled 0-100) (Parikh
185 et al., 2020; Papineni et al., 2002).

186 ▦ MATH: Solve an elementary math story problem by carrying out each arithmetic step and returning
187 the numeric result. Simulates day-to-day problems LLMs may be tasked with by users. GSM8K
188 problems were used and scored by exact match (Cobbe et al., 2021).

### 4.3 Metric Selection

We assess LLM performance in multi-turn tasks by repeating simulations for each instruction and collecting success scores from multiple runs, following Laban et al. (2025). Each score, ranging from 0 to 100, reflects task success. More detailed information on metrics is available in Appendix E

### 4.4 Per-Run Scoring

**I. Binary-Correctness Tasks (Code, Database, API, Math):** A correct response at any turn yields a score of 100, and the run ends. Otherwise, the score is 0.

**II. Refinement Task (Data-to-Text):** The final output is evaluated using BLEU, rescaled to 0–100.

### 4.5 Aggregate Metrics

From the scores collected across the 3 runs, we compute three metrics:

- **Average Performance** ($\bar{P}$)**:** Average performance per instruction for a given task.
- **Aptitude** ($A^{90}$)**:** 90th-percentile score, measures a model's peak capability, indicating its potential to deliver high-quality results in critical multi-turn tasks. Averaged across all tasks.
- **Unreliability** ($U_{10}^{90}$)**:** Difference between 90th and 10th percentiles, quantifies response variability, where lower values reflect greater consistency, essential for user trust and system reliability in long-horizon interactions. Averaged across all tasks.

## 5 Results & Discussion

### 5.1 Average Performance Gains

| Model | 🔁 Code | | | 🗄 Database | | | 🎛 Actions | | | 📝 Data-to-Text | | | ▦ Math | | |
|---|---|---|---|---|---|---|---|---|---|---|---|---|---|---|---|
| | 📄 | ☘ | ↻ | 📄 | ☘ | ↻ | 📄 | ☘ | ↻ | 📄 | ☘ | ↻ | 📄 | ☘ | ↻ |
| ∞ Llama3.1-8b | 21.2 | 21.7 | **52.0**↑ | 47.7 | 25.9 | **64.3**↑ | **83.0** | 45.5 | 60.0↑ | **15.7** | 13.3 | 12.3↓ | 62.6 | 37.4 | **65.7**↑ |
| 🌀 4o-mini | **66.7** | 50.3 | **66.7**↑ | 90.7 | 40.2 | **93.3**↑ | **92.2** | 52.4 | 92.0↑ | **31.2** | 19.8 | 22.0↑ | **88.0** | 58.7 | 85.0↑ |
| 🪟 Phi-4 | 48.4 | 39.1 | **55.0**↑ | **79.6** | 33.1 | 62.0↑ | **76.0** | 34.1 | 65.7↑ | **28.6** | 23.2 | 28.0↑ | **90.4** | 52.5 | 85.3↑ |
| 🌀 4.1 | **88.7** | 72.6 | 81.7↑ | 86.5 | 46.0 | **96.0**↑ | **98.5** | 62.9 | 84.7↑ | **54.4** | 28.6 | 31.0↑ | 89.7 | 70.7 | **91.7**↑ |
| 🌀 4o | **82.9** | 61.3 | 76.3↑ | 91.7 | 42.3 | **95.7**↑ | **97.1** | 65.0 | 82.0↑ | **32.2** | 20.5 | 27.0↑ | **91.9** | 67.9 | 89.3↑ |

Table 1: Average Performance ($\bar{P}$) comparison across three settings: 📄 **FULL** (single-turn), ☘ **SHARDED** (multi-turn baseline), and ↻ **ERGO** (multi-turn with entropy-guided resetting). Arrow represents change in performance for ↻ relative to ☘, with arrow size representing magnitude of change.

Table 1 shows that ERGO delivers substantial performance improvements across all models compared to baseline multi-turn setups. By detecting moments of confusion and restarting interactions, models avoid becoming "lost" in conversational flow. Nearly every dataset and model combination shows increased average success rates, with performance improving by **56.6%** on average and several model-task combinations achieving over **100%** gains compared to original multi-turn baselines.

While FULL is considered our performance upper-bound, ERGO frequently exceeded FULL in both average performance and aptitude (Section 5.2) as our method only corrects derailment when calculated confusion rises significantly. This preserves the model's ability to iteratively reason

and refine responses across shards while preventing the compounding errors typical in prolonged multi-turn contexts. This approach effectively merges both paradigms' strengths: single-turn stability and clarity when needed, and iterative decompositional reasoning when the model remains on track.

Moreover, performance on the 🖼️Data-to-Text task improves over the multi-turn baseline, though less substantially than in other datasets. This is partly due to model-specific constraints. **LLaMA 3.1–8B** struggles to rewrite large, structured prompts effectively (e.g., full tables), limiting the benefit of consolidation. **GPT models** face difficulties in triggering resets, as entropy estimates are less reliable, only top-20 log-probabilities are available, and outputs are typically short, reducing entropy sensitivity. **Phi-4** performs best, nearing single-turn levels, likely because it supports accurate entropy tracking and handles prompt rewriting more effectively. These results indicate model-dependent limitations in applying our method to high input structure tasks.

## 5.2 Aptitude and Unreliability Improvements

Along with performance gains, Figure 3 shows that ERGO demonstrates exceptional gains in aptitude, often exceeding single-turn performance levels, while substantially reducing unreliability compared to multi-turn baselines, two metrics introduced by Laban et al. (2025) to capture model consistency across conversations. These results indicate that our intervention not only fully recovers the aptitude lost in the transition from single-turn to multi-turn settings and achieves aptitude levels exceeding single-turn baselines, but also makes behavior significantly more stable compared to baseline multi-turn settings across repeated trials. When comparing to standard sharded conversations, the average aptitude across models rose by 24.7%, achieving performance levels that surpass single-turn baselines, enabling more effective handling of complex tasks while unreliability declined by 35.3%.

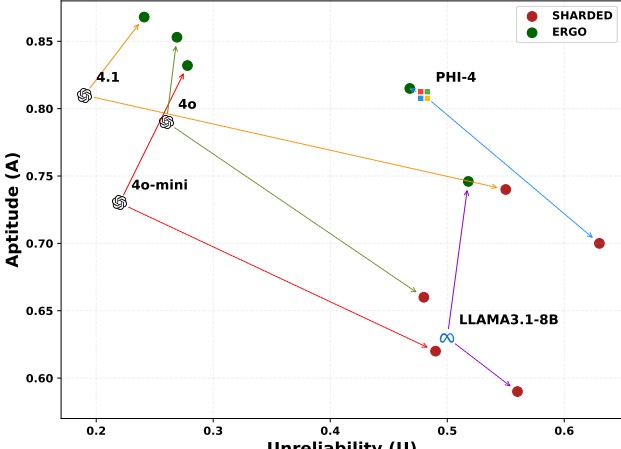

Figure 3: Effect of SHARDED and ERGO on Aptitude and Unreliability. Icons represent models 📄FULL performance. Green dots represent performance with ↻ERGO while red dots represent 🌱SHARDED performance

## 5.3 Evaluating Entropy-Guided Resets vs. Random Resets and Fixed Resets

We compared entropy-based context resets against random and fixed-interval baselines using `Llama3.1-8B` across three tasks: 🗄️*Database*, 🎛️*Actions*, and ⊞*Math*. In these ablations, we retained all experimental settings from the main condition, with the only change being that each metric was tested on 50 question samples instead of 100. The random baseline used uniformly random triggers with unconstrained reset frequency. The fixed baseline triggered resets every five shards (quintet reset), matching the average reset frequency of `Llama3.1-8B` observed in our ERGO system.

The results, visualized in Figure 4, demonstrate a clear advantage for ERGO over baseline approaches. Entropy-guided resets consistently outperformed both random and fixed reset strategies while demonstrating adaptive scaling behavior. In the Database task, ERGO achieved a performance gain vs

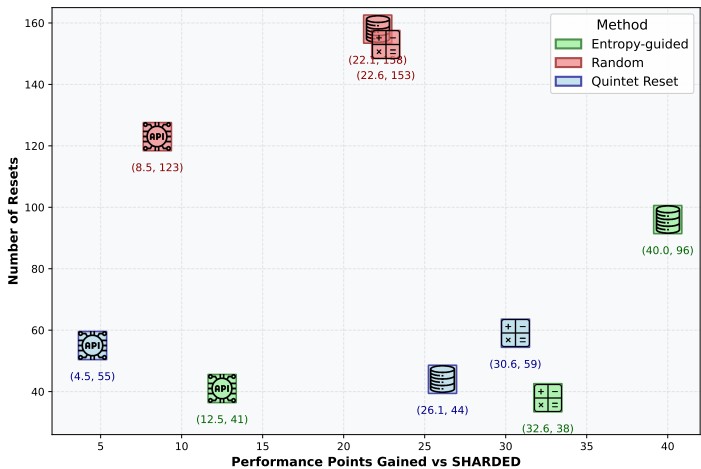

Figure 4: Comparison of performance point gains (percentage-point increase in accuracy relative to 🌿SHARDED) and number of resets across entropy-guided, random, and quintet reset methods on 🗃Database, 🎛Actions, and ⊞Math tasks. Icons represent their respective task with their color determining method used.

SHARDED of 40.0 percentage points using 96 resets, compared to the quintet baseline's 26.1 gain with only 44 resets. This demonstrates the system's ability to increase intervention frequency when encountering greater model uncertainty. Conversely, in the Actions task, ERGO required only 41 resets, fewer than both baselines, while still achieving superior performance. This adaptive behavior indicates that entropy guided resets effectively allocate computational resources by intervening only when necessary, scaling both up and down based on task complexity and model confusion levels.

The primary risk posed by resets is semantic drift. Poorly timed or excessive context rewriting can lose critical details through increased abstraction, compromising semantic faithfulness to the original input (Dreyer et al., 2023). This degradation in semantic faithfulness can offset or even negate the benefits of resetting. Furthermore, resets incur computational overhead; each reset involves having two additional forward passes through the model. Together, these considerations underscore why ERGO outperforms both methods as the frequency and timing of resets are more carefully controlled in our framework. Not only to avoid wasted computation, but, more critically, to prevent semantic degradation. For more information on computation and average reset frequency across models, please refer to Appendix C.

## 5.4 Comparison to Existing Intervention Strategies

To contextualize ERGO's improvements as a novel conversational intervention system, we compare against existing prompt engineering approaches from Laban et al. (2025): **SNOWBALL** and **RECAP** as, to our knowledge, no other methods exist that perform comparable inference time conversational restructuring

⟲ SNOWBALL: Reiterates all prior shards at each new turn, effectively growing the prompt cumulatively.

📖 RECAP: Reiterates all prior shards only at the final turn. While more efficient, this strategy is impractical in real-world deployments, since the system would not know *prior* when the final user input will occur.

| Model | 📄 FULL | 🌿SHARDED | ⟲ SNOWBALL | 📖 RECAP | ⟳ ERGO |
|---|---|---|---|---|---|
| GPT-4o-mini | 73.8 | 44.3 | 54.0 | 57.7 | **71.8** |
| GPT-4o | 79.2 | 51.4 | 57.4 | 66.3 | **75.6** |

Table 2: Comparison of combined average performance ($\bar{P}$) across ⊕Code, 🗃Database, 🎛Actions, 📝Data-to-Text and ⊞Math tasks.

As shown in Table 2, ERGO significantly outperforms both alternatives. ERGO nearly matches single-turn performance for both models. Furthermore, ERGO prevents input bloating at each iteration unlike SNOWBALL, and operates without requiring prior knowledge of the final input unlike RECAP.

## 5.5 Evaluating Length Bias in Entropy-Based Reset Triggers

One potential concern regarding ERGO's entropy-based reset mechanism is whether it inadvertently functions as a proxy for response length. Specifically, since entropy is calculated over token probability distributions, it is plausible that longer outputs, which involve more tokens and potentially more diffuse distributions, may naturally exhibit higher entropy. If true, this would raise the possibility that ERGO's resets are effectively triggered by length increases rather than genuine uncertainty spikes, undermining the validity of entropy as an internal behavioral signal.

We analyze response behavior from the Phi-4 model across all tasks and questions used in the main evaluation suite. For each turn $t$ in a given multi-turn conversation, we compute two quantities relative to the previous turn: the change in average token-level entropy, $\Delta \bar{H}(t)$, and the change in response length, $\Delta L(t)$, measured in tokens.

We evaluate the relationship between these using two standard correlation metrics: Spearman's rank correlation coefficient ($\rho$), which captures monotonic associations without assuming linearity (Spearman, 1904), and Pearson's correlation coefficient ($r$), which quantifies the strength of linear correlation (Pearson, 1895). The results for the Phi-4 model are summarized in Table 3.

The Spearman result indicates no meaningful monotonic relationship between changes in entropy and length. The Pearson coefficient, while statistically significant due to the large sample size, has negligible magnitude and a negative sign, indicating no positive linear correlation.

|  | Coefficient | p-value |
| --- | --- | --- |
| Spearman's $\rho$ | $-0.0143$ | $0.4525$ |
| Pearson's $r$ | $-0.0796$ | $2.7 \times 10^{-5}$ |

Table 3: Correlation between changes in entropy and response length for the Phi-4 model.

These findings demonstrate that entropy fluctuations are not systematically associated with output length changes in the Phi-4 model. This supports the claim that ERGO's reset mechanism is not driven by verbosity or token count, but rather by internal signals of model uncertainty. Entropy-based resets therefore retain validity as an independent control signal rather than acting as a surrogate for response length.

## 6 Conclusion

Our results demonstrate that ERGO provides an effective solution to reliability challenges in interactive ML systems by using Shannon entropy to detect system degradation and trigger automatic context reconstruction. Shannon entropy, despite its computational simplicity, serves as a reliable and precise signal for when interactive systems require intervention to maintain robustness, enabling targeted restoration while minimizing unnecessary computational overhead. ERGO consistently outperformed existing methods, achieving 56.6% performance gains over standard baselines, improving aptitude by 24.7%, and reducing unreliability by 36.3%. ERGO offers a practical, model-agnostic framework for maintaining reliable performance in real-world interactive ML deployments where accumulated context progressively corrupts system behavior. The success of entropy-guided reliability monitoring establishes a new paradigm for robust interactive systems, rather than attempting to prevent degradation, systems can monitor their own reliability in real-time and intervene when uncertainty signals indicate potential failure modes. Future work will explore advanced context consolidation strategies, including multi-stage summarization and adaptive techniques for long-form conversations. More information on Future Works can be found in Appendix D.

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

# A Threshold Selection Procedure

| Model Name | Version | $\tau$ | Percentile | Provider |
|---|---|---|---|---|
| Phi-4 | N/A | 0.1 | 90th | HuggingFace |
| Llama3.1-8b | N/A | 0.03 | 65th | HuggingFace |
| GPT-4.1 | gpt-4.1-2025-04-14 | 0.2 | 90th | OpenAI API |
| GPT-4o-mini | gpt-4o-mini-2024-07-18 | 0.2 | 85th | OpenAI API |
| GPT-4o | gpt-4o-2024-08-06 | 0.3 | 90th | OpenAI API |

Table 4: Model versions, thresholds, and calibration percentiles used in our experiments. (Versions included where applicable.)

To determine appropriate entropy thresholds ($\tau$) for triggering context resets, we conducted a calibration procedure specific to each model. The goal was to identify a rise in entropy that reliably signals when a model is 'lost' in the conversation, that is, when its internal uncertainty increases sharply, suggesting that it is struggling to integrate or reason over the accumulated context.

For each model, we selected a held-out subset of approximately 80 shard-level examples from the GSM8K dataset. These examples were drawn from outside the final evaluation set to avoid contamination, with GSM8K being chosen due to its hybrid structure, requiring both reasoning and natural language generation. We then ran each model in a standard multi-turn setting over these shards and computed the change in average token-level predictive entropy at each turn.

From the resulting distribution of entropy rises, we selected a threshold based on a percentile aligned with the model's baseline aptitude on GSM8K. For instance, since GPT-4.1 achieves a baseline aptitude of $\sim 90\%$ on GSM8K in single-turn settings, we selected the 90th percentile of the entropy rise distribution as its reset threshold. The underlying rationale was to calibrate the threshold so that only the most atypical (high-entropy) turns, those statistically associated with likely failure, would trigger an intervention. Details of the models used, including their version identifiers, selected entropy thresholds, and corresponding calibration percentiles, are summarized in Table 4.

Once determined, this threshold was fixed across all datasets for a given model. We made this decision intentionally, as our goal was to evaluate the feasibility of a general-purpose, model-specific threshold rather than tuning thresholds for each dataset individually. This "one-size-fits-all" approach allows for a more robust and realistic assessment of whether entropy-based context resets can generalize across tasks without requiring per-task adjustment.

Interestingly, while both GPT-4.1 and Phi-4 shared the same 90th percentile threshold, Phi-4 triggered significantly more resets during evaluation. This was due to Phi-4's strong performance on GSM8K but much weaker performance on the broader set of tasks. This divergence illustrates that the system remains sensitive to task-specific confusion, with the number of resets scaling appropriately even under a fixed, model-specific threshold, highlighting the adaptive behavior of the method across domains. More information on number of resets incurred is available in Appendix C

# B   Sensitivity to Entropy Threshold ($\tau$)

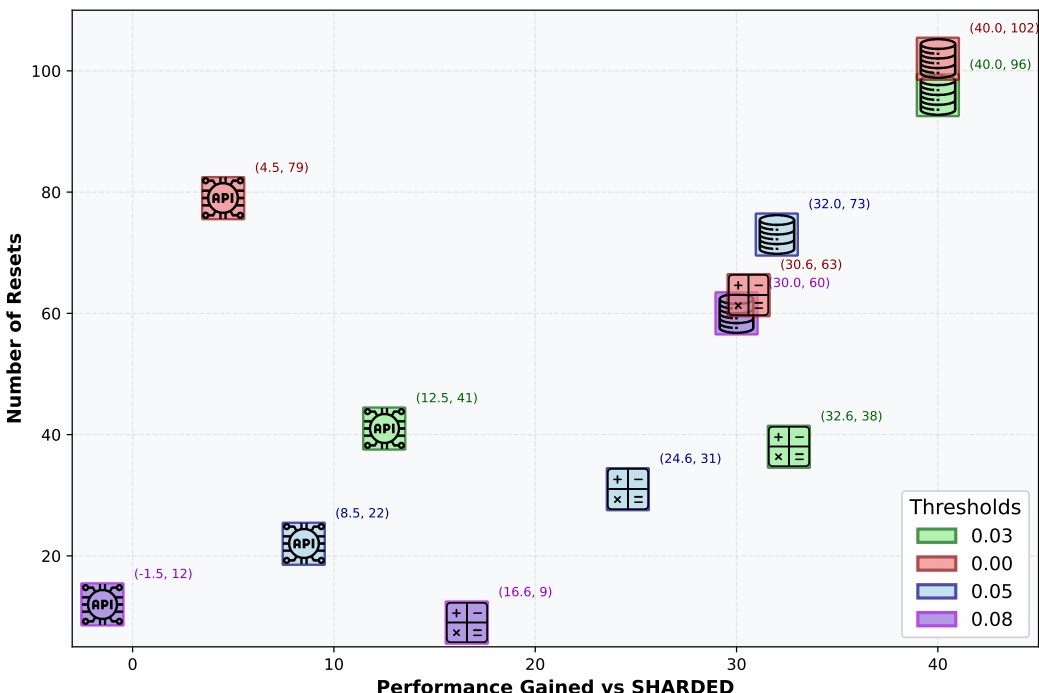

Figure 5: Comparison of maximum performance point gains (i.e., highest percentage-point increase in accuracy relative to 🌿SHARDED) and number of resets between different thresholds on 🗄️ Database, 🎛️ Actions, and 🔢 Math tasks. Icons represent their respective task with their color determining method used.

To evaluate the sensitivity of our method to the entropy threshold parameter $\tau$, we conducted an ablation study using the same controlled setup described in Section 5.3 with the `Llama3.1-8B` model on the *Database*, *Actions*, and *Math* tasks. The only variable changed in this study was the value of $\tau$, the threshold used to trigger entropy-guided resets. We tested four settings: $\tau \in \{0.00, 0.03, 0.05, 0.08\}$, where 0.03 corresponds to the threshold selected for the main experiments.

The results, visualized in Figure 5 showed a clear performance peak at $\tau = 0.03$, which consistently achieved the highest gains across all tasks. This setting struck a balance between reactivity and restraint, triggering resets selectively at moments of genuine confusion without introducing excessive rewrites that risk semantic drift. In contrast, the lowest threshold $\tau = 0.00$ resulted in the highest number of resets and either matched or underperformed the 0.03 setting, suggesting that overly aggressive resetting is not beneficial and may lead to instability due to frequent context rewrites.

At the other extreme, the highest threshold $\tau = 0.08$ yielded the fewest resets and consistently underperformed, likely due to failing to intervene even when the model was demonstrably confused. The intermediate value $\tau = 0.05$ behaved as expected, yielding results that were approximately midpoint between 0.03 and 0.08 in both performance and reset count.

Taken together, these findings support the robustness of our selected threshold and highlight the importance of calibrating reset triggers to maintain a balance between informativeness and intervention overhead.

# C Computational Cost and Reset Overhead Analysis

A key consideration in deploying entropy-guided context resets is the computational overhead they introduce. In our system, two sources of computational cost must be considered: (1) the cost of computing predictive entropy at each turn, and (2) the cost incurred when a context reset is triggered.

**Entropy Computation Cost:** While more advanced measures of model uncertainty such as semantic entropy require sampling multiple outputs over the same input (Kuhn et al., 2023), our method uses token-level Shannon entropy, which is extracted directly from the next-token probability distribution during generation. This choice imposes negligible additional cost beyond standard decoding and was selected for its practicality and compatibility with real-time systems.

**Reset Overhead:** Each reset introduces two additional forward passes through the model: one to rewrite the accumulated user context into a consolidated prompt, and a second to respond to that prompt. This introduces latency and compute proportional to the number of resets triggered per run. Table 5 showcases the average performance of models with ERGO along with the approximate number of shards per reset and the selected threshold percentile for each model. Averaged across all datasets, one question equates to $\sim 6$ shards.

| Model | Average Performance | $\sim$ Shards per Reset | Threshold Percentile |
|---|---|---|---|
| GPT-4o | 75.6 | 51 | 92nd |
| GPT-4.1 | 77.2 | 38 | 90th |
| GPT-4o-mini | 71.8 | 29 | 85th |
| Phi-4 | 59.2 | 7 | 90th |
| Llama3.1–8B | 50.9 | 5 | 63rd |

Table 5: Average Performance with ERGO along with the number of shards before reset occurs for each model and its threshold percentile, measured as an average across all datasets.

These results reflect the adaptive nature of the system: more capable models (e.g., GPT-4.1, GPT-4o) experience fewer high-entropy turns and thus require fewer resets, minimizing overhead. Conversely, less capable models like `Phi-4` trigger resets more frequently, aligning with their observed confusion.

**Prompt Length Reduction:** An additional consequence of context resets is that they tend to truncate the context window, potentially removing stale or redundant information. Across all runs, the average token length of model prompts for questions where resets occurred was 260 tokens, compared to 309 tokens in questions where no resets were triggered. While this reduction does not eliminate the cost of the reset itself, it may partially offset it by reducing input size in subsequent turns.

**Retrieval-Augmented Consolidation (Future Work):** More advanced consolidation techniques, such as retrieval-augmented synthesis, could further improve the quality of resets but would introduce additional retrieval and ranking costs. We leave the exploration of such hybrid architectures to future work.

Taken together, these results indicate that while entropy-guided resets do introduce compute overhead via additional forward passes, the system remains adaptive. Reset frequency scales with model confusion, and thresholds derived from a single reasoning heavy dataset generalize effectively across diverse tasks.

## D  Future Works

While ERGO has demonstrated substantial improvements in multi-turn performance through entropy-guided context resets, several avenues remain open to extend its applicability and robustness in broader conversational settings.

**Dialogue Trace Consolidation:** Our current context-reset protocol rewrites prior *user* inputs into a single-turn prompt but does not incorporate preceding *assistant* responses. This simplification was chosen to enable stateless resets with minimal overhead in instruction-shard tasks, where user inputs encode the majority of required task information. However, in more open-ended or exploratory conversations, where assistant turns may introduce novel entities, explanations, or intermediate reasoning, this exclusion could result in loss of critical context.

To address this, future work will explore **multi-stage consolidation mechanisms** that explicitly summarize both user and assistant dialogue turns. One natural extension is a two-pass strategy: the first pass summarizes user queries, and the second distills assistant responses. A final generation step would synthesize these into a coherent prompt, preserving key semantic and referential content across turns. This approach maintains ERGO's core design, resetting when confusion is detected via internal uncertainty signals, while enhancing its fidelity in dialogic settings.

**Adaptive Consolidation Strategies:** Incorporating assistant responses also raises new design challenges around content selection, co reference resolution, and context prioritization. We anticipate integrating lightweight discourse-aware filtering or retrieval-augmented synthesis to further improve semantic coverage without incurring significant computational cost. Evaluating these techniques on long-form conversations, assistance tasks, and real-world dialogue logs will be a focus of future iterations.

These extensions do not alter the core entropy-based mechanism but instead refine how reset inputs are constructed. As such, they represent a natural progression of ERGO's architecture toward more general-purpose deployment. Further exploration will also include model-internal dynamics beyond entropy, adaptive thresholding tuned to conversation domain, and integration with memory or retrieval components to support resets over extended dialogue spans.

## E  Metrics

### E.1  Metric Selection

LLMs employ a stochastic decoding process, yielding different outputs even under fixed prompts and sampling parameters. We leverage this by repeating our multi-turn simulation on each sharded instruction and observing the resulting success scores. Let

$$S = \{ S_i \}_{i=1}^{N}$$

be the set of scores from $N$ independent runs on a single instruction, where each $S_i \in [0, 100]$ measures task success at the end of that simulation.

#### E.1.1  Per-run scoring:

  I. **Binary-correctness tasks (Code, Database, API, Math):** At each turn, we evaluate the model's response; if it produces a correct solution at any turn, we immediately assign $S_i = 100$ and terminate that run. If no turn yields a correct answer, $S_i = 0$.

 II. **Refinement task (Data-to-Text):** We compute the native metric (BLEU for data-to-text; joint coverage/attribution score for summarization) on the final generated output and rescale it to $[0, 100]$.

#### E.1.2  Aggregate metrics

From the per-run scores $S$, we define three summary statistics, following the methodology from Laban et al. (2025):

$$\bar{P} = \frac{1}{N} \sum_{i=1}^{N} S_i \tag{1}$$

$$A^{90} = \text{percentile}_{90}(S) \tag{2}$$

$$U_{10}^{90} = \text{percentile}_{90}(S) - \text{percentile}_{10}(S) \tag{3}$$

- $\bar{P}$ (Average Performance): An unbiased estimate of the model's mean score on an instruction.

- $A^{90}$ (Aptitude): Estimates the 90th-percentile performance, reflecting what one can achieve in the top decile of runs.

- $U^{90}$ (Unreliability): Measures the gap between the 90th and 10th percentiles, capturing the degree of stochastic variability in outputs.

Aptitude and Unreliability are computed per instruction and then averaged over the full set of tasks. Binary-correctness accuracy is mapped onto the 0–100 scale to ensure every task's score aligns.

