# OpenReview forum: "ERGO: Entropy-guided Resetting for Generation Optimization in Multi-turn Language Models"
_NeurIPS.cc/2025/Workshop/Reliable_ML — NeurIPS 2025 - Reliable ML Workshop_

### Official Review · Reviewer_1xd2 · 2025-09-13
**Strong paper proposing a method to improve reliability in multi-turn language models.**

**Rating:** 7
**Confidence:** 3

**Review:**

1. This paper presents a new method to improve the reliability of multi-turn language models. This model involves tracking the entropy from one turn to the next. If the entropy is ever too high, the method "resets" the conversation by summarizing the user's prompts across the past turns into a single prompt and feeding it into a fresh copy of the model. The authors show strong empirical performance of this method across a wide range of tasks and base LLMs, often outperforming baselines and existing methods.

2. This paper is really well-written. I especially really liked Lines 93-98, which I thought did a good job at contextualizing their contributions against existing approaches. I also thought the proposed method was simple, natural, and empirically promising.

3/4. I don't have any major weaknesses. That said, I think the authors can do a better job at providing more details about how one should pick \tau in practice. In addition, I also wonder whether \tau can be picked adaptively instead of being fixed. Finally, I think it would be interesting to see how your method compares to existing ones in terms of wall-clock latency or FLOPs.

---

### Official Review · Reviewer_f1Yg · 2025-09-15
**A good paper on enhancing reliablity of long multi-turn conversations**

**Rating:** 7
**Confidence:** 2

**Review:**

Summary: This paper aims to tackle the problem of significant performance degradation of LLMs in multi-turn environments. They propose an approach called ERGO (Entropy-guided Resetting for Generation Optimization) that maintains system reliability and performance. Specifically, this approach contains two components: (1) a metric that monitors internal uncertainty, which is the change of the averaged Shannon entropy of the next-token probability distribution in each turn, and (2) when the change is above a threshold, indicating potential failure, another LLM will generate a summary of the context and reset the conversation. This paper includes fairly extensive experimental results to demonstrate the strong performance of ERGO over degraded multi-turn baselines and existing intervention strategies, as well as ablation studies on random/fixed resets and length bias.

I think this paper studies an important question in reliable ML and is relevant to the workshop. This paper is well-written and easy to follow. The experiments are extensive. The idea of using the entropy of internal token distribution as a signal for model uncertainty/unreliability seems natural, but the authors argue that they are the first practical framework for maintaining reliability and performance in interactive ML environments through dynamic uncertainty monitoring. This seems to be a strong claim that requires more context to verify.